# Is health literacy of family carers associated with carer burden, quality of life, and time spent on informal care for older persons living with dementia?

Kristin Häikiö[1,2☯]*, Denise Cloutier[3‡], Jorun Rugkåsa[1,4☯]

1 HØKH–Health Services Research Unit, Akershus University Hospital, Lørenskog, Norway, 2 Department of Nursing and Health Promotion, Faculty of Health Science, Oslo Metropolitan University, Oslo, Norway, 3 Department of Geography & Institute on Aging & Lifelong Health, University of Victoria, Victoria, British Columbia, Canada, 4 Center for Care Research, University of South-Eastern Norway, Porsgrunn, Norway

☯ These authors contributed equally to this work.
‡ These authors also contributed equally to this work.
* haikio@oslomet.no

**Data Availability Statement:** The full dataset cannot be shared publicly because of risk of identification. A minmal dataset is provided so that our analysis and results can be replicated.

## Abstract

### Introduction

Family carers are cornerstones in the care of older people living with dementia. Family carers report extensive carer burden, reduced health-related quality of life (HRQoL), and extensive time spent on informal care (Time). Health literacy (HL) is a concept associated with people's ability to access health services, and navigate the healthcare system. This study's aim was to investigate HL among family carers, and investigate the associations between HL and carer burden, HRQoL, and Time spent on informal care.

### Method

We designed a self-administered survey comprising validated instruments, including the Health Literacy Scale (HLS-N-Q12) to measure HL, Relative Stress Scale (RSS) to measure carer burden, the EQ-5D-5L instrument to measure HRQoL, and some modified questions from the Resource Utilization in Dementia (RUD) questionnaire to measure time spent on informal care (Time). Descriptive analysis in addition to bivariate and multiple linear regressions were undertaken. In multiple linear regression analysis, we used HL as the independent variable to predict the outcomes (carer burden, HRQoL, Time). Analyses were adjusted for the effects of explanatory independent variables: age, gender, education levels, urban residency, having worked as health personnel, caring for someone with severe/mild dementia, and being born abroad.

### Findings

In a non-probability sample of 188 family carers from across Norway, most of them female and over the age of 60, we found high levels of HL. In the bivariate analysis, carer burden and HRQoL (EQvalue) showed significant associations with HL. In the multiple regression

**Funding:** JR received funding to this Project (ref. nr. 256431) by the Norwegian Research Council, https://www.forskningsradet.no/en/. The funders had no role in study design, data collection and analysis, decision to publish, or preparation of the manuscript.

**Competing interests:** The authors have declared that no competing interests exist.

analyses, HL was statistically significantly associated with carer burden (B = -0.18 CI:-0.33,-0.02 p = 0.02), HRQoL (EQvalue: B = 0.003 with 95% CI: 0.001, 0.006 p = 0.04), and Time (B = -0.03 with 95% CI: -0.06, 0.000, p = 0.046), after adjusting for the effect of independent variables.

## Conclusion

This is one of the first studies to investigate the associations between HL and different outcomes for family carers of older people living with dementia. Additional research into the associations identified here is needed to further develop our understanding of how to support family carers in their roles. Targeted support that increases family carers' HL may have potential to enhance their ability to provide sustainable care over time.

## Introduction

The concept of Health literacy (HL) relates to self-management and constructive communication in healthcare contexts and concerns someone's ability to utilize health information to make decisions about their health. HL has been defined in many ways, but based on a systematic review of different definitions, it can be described as:

> "*people's knowledge, motivation and competencies to access, understand, appraise, and apply health information in order to make judgments and take decisions in everyday life concerning healthcare, disease prevention and health promotion to maintain or improve quality of life during the life course.*" [1, p. 3]

Understood in this way, HL may be a key factor in how people manage their health [2]. Since the level of HL will vary across individuals in a population; it might be one factor contributing to unequal access to health services, and inequalities in health outcomes [3, 4]. For example, low levels of HL have been found to increase the use of acute care health services and prolong hospital stays [5]. Like literacy in general, HL is thought to be a learned skill [6], and it is positively associated with level of education in many studies [5]. A systematic review aimed at establishing the efficacy of interventions to improve health literacy and health behavior, concluded that HL interventions hold potential to combat health inequalities [2, 7, 8].

Studies of HL often focus on specific groups of patients. More recently, some studies have also focused on HL among family carers. Of the studies that have investigated HL and family carers, some have found that increased HL among family carers' improves knowledge about existing health services and the importance of keeping up to date about services [9], and improves communication with health personnel [10]. Higher levels of HL have also been associated with better access to carer support services [2–5].

In international and Norwegian health policy, family carers play an essential role in enhancing quality of care, supporting access and utilization of necessary services, and maintaining good quality of life through care that is tailored to individual needs [11, 12]. Family carer roles may be particularly important in the care of those with dementia, who often experience behavioral and cognitive challenges [13, 14], and progressive disability and dependency [15]. Family carers will often be the ones to notice symptoms first [16], and who, as the illness progresses, take on increasing care responsibilities [17]. Therefore, throughout the illness trajectory, family carers often play a key role in finding and accessing information and services for people

living with dementia [18]. Consequently, their level of HL might thus potentially impact the services received [19, 20].

Given the rising number of older persons living with dementia world-wide [21], and the extensive role of family carers in their care, there is good reason to investigate how HL might be associated with the ability of family carers to carry out their role in the longer term. In this article, the care provided by family carers is referred to as informal care, in contrast to formal care which is care delivered by health professionals.

In the literature, it is well known that family carers of people living with dementia, experience heavier care burdens and poorer quality of life as a result of the informal care that they provide compared with those caring for family member with other conditions [22, 23]. The time spent on informal care can be considerable [24] and may limit the ability of carers to take part in recreational or social activities [25]. It might be the case that family carers with high levels of HL are better able to obtain support services that can mitigate some of these negative outcomes [19, 20, 26].

It makes sense that improved access to health services for the person living with dementia, gained through better knowledge about services and improved communication might also reduce carer burden, as well as increase quality of life for individuals and carers, and relieve the carer from tasks that they would otherwise do [27, 28]. For this reason, it is of interest to explore associations between the HL of family carers, and outcomes, such as carer burden, health-related quality of life (HRQoL), and the time spent on informal care. Very limited research exists, however that has examined these relationships.

A recent scoping review examined the relationship between HL and health outcomes for family carers [26]. The authors found one study that had investigated the association specifically between HL of family carers and family carers' outcomes indicating that low HL among family caregivers of older people, may increase carer burden [26]. The authors of the review emphasized the need for further examination of these associations. We were unable to find any studies that specifically examined HL in relation to key outcomes such as carer burden, health-related quality of life (HRQoL), and the time spent on informal care for family carers of older people with dementia. This is the gap that this paper aims to fill.

In the next section, we report findings from a survey of family carers from across Norway, designed to improve our understanding of the role that health literacy (HL) plays in family caregiving. The aims of this article are, first, to describe the level of HL among family carers of older people living with dementia, and second, to investigate whether there are associations between HL and: a) carer burden, b) health-related quality of life and c) time spent on informal care.

## Methods

We constructed a survey to measure HL and the mentioned outcomes among adult family carers of older person living with dementia across Norway.

### Study participants and recruitment strategy

We define a family carer as someone (a family member, neighbor or friend) who, due to the care recipient's health situation, carries out tasks of a supportive nature that go beyond normal relationships of reciprocity among adults [29]. Persons eligible for the study included any family carer to a person above 65 years of age with suspected, or diagnosed dementia, or symptoms of age-related memory loss. The care recipient could be living independently, in a nursing home facility, or in the same household as the carer.

We obtained a non-probability, opportunistic, convenience sample of family carers by contacting a large number of health personnel, such as those working in local dementia teams, out-patient clinics, nursing homes, and home care service providers, to help with recruitment. They were asked to distribute paper versions of the survey questionnaire, as well as to share a link to the electronic version. They also distributed a one-page information sheet that included a Quick Response code (QR code) that linked to the electronic version of the survey, and were asked to share the electronic link on their web pages or in relevant social media groups. We distributed 410 paper surveys, and 235 one-page information sheets to health personnel for redistribution to family carers. We have no information on the number of family carers who were exposed to information about the survey.

Those who responded online had to provide consent before opening the survey. The information sheet enclosed with the paper version informed potential participants that we considered receipt of the completed form as their consent to participate. All participants were informed that they could withdraw from the study after submission by contacting the research team. No participant availed themselves of this opportunity.

Data were collected between January—May 2019 and the questionnaire was self-administered. Paper-surveys were returned in a closed, prepaid envelope, while electronic forms were submitted online and forwarded to the research team via an encrypted data server.

The Research Ethics Committee judged this study outside their remit as defined by the Norwegian Health Research Act (ref.nr 2018/1725 C). After assessing our data protection and privacy impact assessment (DPIA) and potential risks for participants, the study was approved by the Akershus University Hospital's Privacy Ombudsman (ref. 2018–126).

## Measures

*Health Literacy (HL)* was measured using the Health Literacy Scale, Norwegian translation (HLS-N-Q12), which is a 12-item, validated scale [30]. Each item is scored on a 6-point Likert scale ranging from 1 = "very difficult" to 6 = "very easy". Summed scores range from 12–72, with higher scores indicating a higher level of HL [31, 32]. To represent the level of HL as descriptive categories, we converted the 6 levels to 4 as suggested by the scale's developers, and we followed their procedures for calculating cut-off values [33]. By merging the four middle categories into two middle categories, and applying the cut-off values, the four levels of HL were identified as: inadequate level (score 12–26), marginal level (score 27–32), intermediate level (score 33–38) and advanced level (score 39–48). The 6-level measure was used in the regression analysis, and the 4-level measure in the descriptive analyses.

*Carer burden* was measured using the Relative Stress Scale (RSS), which has been used to measure such burden in previous dementia research in Norway [34]. RSS measures subjective burden in three areas: 1) emotional distress, 2) social distress, and 3) negative feelings [34]. Each of 15-items is scored on a 4-point Likert scale from 0 = never to 4 = always. Summed scores range from 0–60, and higher scores indicate higher levels of carer burden.

*Health Related Quality of Life* (HRQoL) was measured using the Norwegian translation of the EQ-5D-5L [35]. This instrument yields two values for HRQoL: a health profile (*EQvalue*), and a visual analogue scale (*EQvas*). The EQvalue is calculated by asking participants to rate potential problems in five dimensions of health: mobility, self-care, usual activities, pain/discomfort, and anxiety/depression. Each is rated on a five-point scale with levels of problems ranging from 1 = no, 2 = slight, 3 = moderate, 4 = severe, and 5 = extreme [36]. Each of 3125 possible combinations of responses (health states) is assigned a health state value, the EQvalue, and reported on a scale where 1 is equivalent to full health, and 0 is equivalent to being dead [36]. The EQvas was measured with a vertical, thermometer-like visual analogue scale (VAS),

where respondents rated their current health on a scale ranging from 0 = "worst imaginable health" to 100 = "best imaginable health" [36].

*Time spent on informal care (Time)* was measured using items from the Resource Utilization in Dementia Questionnaire (RUD) [37], adjusted to fit the research questions. This included adding one more traditionally male chore; maintenance of the house, to make the measure more relevant to male carers. We also added time spent talking to the participant on the phone and time spent interacting with health personnel. Tasks were grouped in clusters: 1 = personal care, 2 = gardening, house work, shopping, medication and economy, 3 = talking with the care recipient on the phone, 4 = attending appointments with the care recipient, 5 = interacting with health personnel, or searching for information about services. The time spent on each of the five task clusters were calculated by multiplying the "hours spent on a typical care day" with the "number of days" in which this task was carried out during the last 30 days. The Time variable is calculated by adding up the time spent on all task clusters.

*Dementia severity* was measured using the Norwegian translation of the Berger Dementia Scale (BDS) to distinguish between those caring for persons living with severe versus mild dementia [38, 39]. The BDS was chosen because it is easy to use, does not demand any medical assessment, and has been used in previous research [40, 41]. The BDS consists of 6 statements that describe different levels of functioning, and asks the family carer which level best describes their care recipient. The statements are ordinal with the first three levels of BDS being classified as "mild dementia" (= 0) and the remaining three as "severe dementia" (= 1), as advised in the literature [39]. We assumed that differences in dementia severity could affect the carers' outcomes and we wanted to be able to adjust the regression analysis to account for this potential effect.

Socio-demographic variables included *age* (in years), and *gender* (female = 0/male = 1). *Urban residency* was coded on the basis of respondents' postal code (rural = 0/urban = 1). *Highest education achieved* was collected as 1 = primary school (9 years), 2 = secondary school (12 years), 3 = Up to three years of university education, and 4 = more than three years of university education. For use in the regression analyses, we distinguished between lower level of education (primary and secondary school, and up to 3 years of university education) and higher level of education (> 3 years of university education), and converted the variable into a dichotomous variable *higher education* (no = 0/yes = 1). *Health personnel* was a dichotomous variable, indicating whether the respondent had ever worked as health personnel (no = 0/yes = 1). The variable was included as work experience in the health sector might be linked with higher levels of HL. *Carer born abroad* (no = 0/yes = 1) indicates whether the carer was born outside of Norway, and was included because earlier studies have shown that immigrants may have lower HL than the other groups [20]. Information about whether the care recipient was living in a nursing home facility or not was collected and the variable was coded as: living in a nursing home facility (no = 0/yes = 1).

## Statistical analysis

All analyses were conducted using SPSS version 25 for Windows. All tests were two-sided. Results with p-values below 0.05 were considered statistically significant.

To address our first research aim, we conducted descriptive analyses. We report categorical variables using percentages (%) and the number (n) of valid participants, and the number of missing values. For all continuous variables, we report the median, minimum and maximum values. The median was chosen instead of the mean because the distributions of several variables were skewed, and the median is a better measure of central tendency for skewed data. We chose to present all continuous variables in the same way.

To meet our second research aim, we conducted regression analyses in which the main independent variable was HL and the outcome variables of specific interest were carer burden, health-related quality of life (EQvalue, and EQvas) and Time.

We used bivariate regression analyses to investigate the association between HL and the outcome variables. We then used multiple regression analyses to investigate linear associations between HL and the four outcome variables when adjusting for the effect of the following 8 pre-specified explanatory dependent variables: age, gender, higher education, urban, health personnel, dementia severity, and carer born abroad. The explanatory variables were chosen based on theory, correlation analysis, and the number of cases in the dataset. Imputation was undertaken for missing values for the following three of the variables: HL, carer burden and Time.

For the HL-variable and the carer burden variable; missing values were replaced with a mean value for each case [42, 43]. In those cases were imputation was undertaken, fewer than 3 values were missing.

For the Time variable imputation was undertaken according to the following two procedures: 1) In the few cases where the respondent had answered only one of the two questions used to calculate time spent on informal care, either "hours spent", or "number of days", we used imputation of the mean value for the sample. 2) When both values in a cluster of tasks were missing we used the value zero. This occurred in 32 cases. While this included the risk of underestimating the time spent on informal care, it reduced the risk of Type 1 errors in the regression analysis. To test whether this procedure increased the risk of Type II errors, we conducted a sensitivity analysis with imputed mean values.

We investigated the associations between HL and each of the outcome variables in the four multiple regressions models: Model 1: The relationship between HL and carer burden; Model 2: The relationship between HL and EQvalue; Model 3: The relationship between HL and EQvas; Model 4: The relationship between HL and Time.

HL, carer burden, EQvalue, and EQvas were treated as continuous variables in the regression models, using 6-levels of HL.

Model assumptions for the multiple regression analyses were tested. All variables except the Time variable had normally distributed residuals. We therefore used log time transformation [42] to meet the assumptions of linearity of the residuals for the Time variable. One minute was added to the total time for each participant in order to avoid logarithms of zero for these calculations. Linearity was checked using scatterplots and correlation analyses. The linearity between HL and the four outcome variables (carer burden, EQvalue, EQvas, and Time) was weak, and for that reason we further investigated the relationship between variables with box-plots of the quartiles of each outcome variable and HL. No other relationship than linear was found between the outcome variables and HL. Homoscedasticity was checked visually on scatterplots, and there was no substantial deviation from normality. There were no multicollinearity issues in the models (VIF $< 2$).

## Results

As noted, a total of 188 questionnaires were returned. As shown in Table 1, the majority of participants were females (71%, n = 134), and the median age was 60 years old, ranging from 25 to 84. The majority of participants lived in urban areas 87% (n = 160). Around two-thirds (65%, n = 105) indicated that they were caring for a person with mild dementia. A third of the sample (31%, n = 54) had experience as health personnel. Only 9% of participants (n = 17) were born outside of Norway.

The median and minimum-maximum values for HL and the other outcome variables (carer burden, HRQoL, and Time) are shown in Table 1. The median value of the 6 level HL-

**Table 1. Characteristics of the sample, n = 188.**

|  | Valid | Missing |
|---|---|---|
| **Age, median (min-max)** | 60 (25–84) | 0 |
| **Gender, female, n (%)** | 134 (71) | 1 |
| **Highest educational achievement:** |  | 2 |
| **Primary school (9 years), n (%)** | 11 (6) |  |
| **Secondary school (3 years), n (%)** | 47 (25) |  |
| **University education (1–3 years), n (%)** | 45 (24) |  |
| **Higher university education (>3 years), n (%)** | 83 (44) |  |
| **Living in urban areas** | 160 (87) | 4 |
| **Have worked as health personnel, n (%)** | 59 (31) | 7 |
| **Dementia severity of care recipient:** |  | 1 |
| **Mild dementia, n (%)** | 120 (64) |  |
| **Severe dementia, n (%)** | 67 (36) |  |
| **Carer born abroad, n (%)** | 17 (9) | 2 |
| **Health literacy (HL 6 level scale), median (min-max)** | 61 (12–72) | 8 |
| **Health literacy (HL 4 level scale), median (min-max)** | 41 (12–48) | 8 |
| **Advanced level of HL, n (%)** | 104 (58) |  |
| **Intermediate level of HL, n (%)** | 59 (33) |  |
| **Marginal level of HL, n (%)** | 9 (5) |  |
| **Inadequate level of HL, n (%)** | 8 (4) |  |
| **Carer burden, median (min-max)** | 26 (0–48) | 7 |
| **HRQoL (EQvalue), median (min-max)** | 0.79 (0.09–1) | 5 |
| **HRQoL (EQvas), median (min-max)** | 80 (20–100) | 0 |
| **Time (hours) spent on informal care pr month, median (min-max)** | 52.1 (00–1520*) | 0 |

* Answers exceeded the maximum number of hours per month for 5 cases.

scale was 61 (min-max: 12–72). The four descriptive categories of HL showed that almost two-thirds were at an advanced level (58%; n = 104), one third at intermediate level, and only 9% (n = 17) reported inadequate or marginal levels of HL.

The median score for carer burden was 26 (min-max: 0–48), and the median value for EQvalue was 0.79 (min-max: 0.09–1). For the EQvas the median value was high, at 80 (min-max: 20–100). The median time spent on informal care was 52.1 hours over the previous month (min-max: 0–1520 hours). Five cases reported time spent on care that exceeded 24 hours per day (i.e., >720 hours per month), possibly reflecting that several chores were carried out simultaneously. These five cases were kept in the sample after sensitivity analysis showed that excluding them from the dataset did not significantly change the results of the regression analysis.

Table 2 shows that in the bivariate regression analyses, HL was significantly associated with carer burden (B -3.21 CI:-0.37,0.34 p = 0.01) and the EQvalue (B 0.003, with 95% CI: 0.003,-0.005, p = 0.01).

The results from the multiple regressions are shown in Table 3. When adjusting the regression analyses for the explanatory independent variables, significant associations remained between HL and carer burden (B = -0.18 CI:-0.33,-0.02 p = 0.02), and between HL and EQvalue (B = 0.003 with 95% CI: 0.001, 0.006 p = 0.04). In addition it was significant for HL and Time (B = -0.03 with 95% CI: -0.06, 0.000, p = 0.046).

Of the explanatory independent variables affecting carer burden, being male, being born abroad, and caring for someone with severe dementia were statistically significantly associated

Table 2. Bivariate regression analysis with HL[¶] as independent variable.

|  | B (95% CI) | Sig | N | $R^2$ |
|---|---|---|---|---|
| Carer burden | -0.21 (-0.37, 0.04) | 0.01* | 179 | 0.03 |
| EQvalue | 0.003 (0.001,0.005) | 0.01* | 177 | 0.04 |
| EQvas | 0.21 (-0.06,0.47) | 0.13 | 179 | 0.01 |
| Time | -0.03 (-0.05, 0.003) | 0.08 | 179 | 0.02 |

[¶] The 6-level scale of HL is used in the regression

*$p < 0.05$.

with increased carer burden, while having completed more than 3 years of university education was significantly associated with less carer burden. See Table 3.

Several explanatory independent variables were associated with the outcome measures, as shown in Table 3. Increased EQvalue was associated with being male, and living in urban areas. Being male was also associated with EQvas, as was higher education (>3 years or higher university education). More time spent on informal care was associated with higher age and being female.

## Sensitivity analyses

In 32 cases all values were missing in a cluster for the Time variable. The sensitivity analysis where we used imputed means instead of zero in these cases showed that a higher level of HL was still statistically significantly associated with less time spent on informal care (B -0.33 with 95% CI: -0.07, 0.000, p = 0.049). Five participants indicated having spent time exceeding the number of hours in a month (>720. Therefore, we ran the regression analyses of Time again, excluding these five cases. HL remained statistically significantly associated with Time (B -0.29 with 95% CI: -0.06,0.000, p = 0.049).

Table 3. Multiple linear regression models of HL and the four outcome variables[¶].

|  | Model 1, n = 168 Carer burden $R^2$ = 0.224 | | Model 2, n = 167 Health-related quality of life (EQvalue) $R^2$ = 0.146 | | Model 3, n = 168 Health-related quality of life (EQvas) $R^2$ = 0.121 | | Model 4, n = 168 Time[a] $R^2$ = 0.075 | |
|---|---|---|---|---|---|---|---|---|
|  | B (95% CI) | Sig | B (95% CI) | Sig | B (95% CI) | Sig | B (95% CI) | Sig |
| (Constant) | 33.39 (20.91,45.87) | 0.00 | 0.51 (0.32, 0.69) | 0.00 | 51.89 (30.49,73.29) | 0.00 | 4.68 (2.34,7.04) | 0.00 |
| Health literacy | -0.18 (-0.33,-0.02) | 0.02* | 0.003 (0.001,0.006) | 0.04* | 0.20 (-0.06, 0.46) | 0.13 | -0.03 (-0.58, 0.000) | 0.046* |
| Age | 0.09 (-0.04, 0.22) | 0.18 | -0.001 (-0.002,0.001) | 0.59 | -0.05(-0.27, 0.17) | 0.65 | 0.03 (0.001, 0.05) | 0.04* |
| Gender, male[b] | -8.93 (-12.88, -4.98) | 0.00* | 0.11 (0.05, 0.16) | 0.000* | 8.36 (1.58, 15.13) | 0.02* | -0.92 (-1.67, -0.18) | 0.02* |
| Higher education[b] | -4.76 (-8.37,-1.16) | 0.01* | 0.03 (-0.03, 0.08) | 0.31 | 6.69 (0.51, 12.87) | 0.03* | -0.17 (-0.85, 0.51) | 0.63 |
| Urban residency[b] | 1.42 (-3.73,6.56) | 0.59 | 0.08 (0.01, 0.15) | 0.048* | 8.35 (-0.47, 17.17) | 0.06 | -0.33 (-1.30, 0.64) | 0.50 |
| Health personnel[b] | -1.66 (-5.53, 2.20) | 0.40 | 0.03 (-0.03, 0.09) | 0.26 | 5.56 (-1.06, 12.19) | 0.10 | -0.30 (-1.03, 0.42) | 0.41 |
| Dementia, severity[c] | 5.47 (1.83, 9.11) | 0.003* | -0.004 (-0.06, 0.05) | 0.90 | -1.00 (-7.25, 5.24) | 0.75 | -0.07 (-0.76, 0.62) | 0.84 |
| Carer born abroad[b] | 6.95 (0.71, 13.19) | 0.03* | -0.002 (-0.09, 0.09) | 0.96 | -4.34 (-15.05,6.37) | 0.43 | 0.04 (-1.14, 1.21) | 0.95 |

[¶] Note: The 6-level scale of HL is used in the regression

*$p < 0.05$.

[a] The Time variable is the log transformed variable.

[b] Variables are binary and coded no = 0/yes = 1.

[c] The variable Dementia severity is coded mild dementia = 0/severe dementia = 1.

It is possible that the large differences in time spent on informal care could be related to whether the care recipient lived in a nursing home, which might reduce the need for carer input. We conducted a sensitivity analysis on the time variable using independent sample T-test to investigate difference of mean between those who cared for a person living in a nursing home and the remaining respondents. We found no statistically significant difference (mean 3.6 hours, SD 2.3 vs .2 hours, SD 2.6, p = 0.31). The correlation between the time variable and caring for a person living in nursing home was also not significant (Pearson's R = -0.14, p = 0.051).

## Discussion

In a non-probabilistic sample of family carers of older people living with dementia, we found significant associations between HL and carer burden, HL and health-related quality of life when measured by the EQvalue, and between HL and time spent on care.

### Health literacy among family carers

In general, there was a high level of HL in our sample. As many as 58% were in the category of advanced HL and only a small proportion of participants who had marginal or inadequate levels of HL (9%). This diverges substantially from previous studies of HL. Among people living with diabetes, and in the general population in Norway, 34–41% had low levels of HL (inadequate) [33]. In the general population across Europe, insufficient or problematic levels of HL were reported for 47% of the population [44]. Even though measurement instruments and the definitions of what constitutes low levels of HL differ between studies, our findings indicate a higher level of HL. This could result from participants having acquired HL as part of their experience as family carers [6]. Also, there may be selection bias in the participants in this study, such that those who chose to participate had higher levels of HL than the general population [45, 46]. A self-administered survey makes cognitive demands on participants [45], which might impact who volunteered to take part [46]. It may be likely that the combination of a non-probability sampling method, voluntarily participation of family carers, the self-reported nature of the HL-scale, and self-administration of the survey, have together contributed to participants with higher levels of HL.

Previous research from the US has found that adults without a high school diploma, with health-related restrictions and limited access to resources, who are immigrants and members of other minority populations have lower HL skills than others [47]. These groups were under-represented in our sample. Participants born outside of Norway accounted for only 7% of our sample compared with 14% in the general Norwegian population [48]. A positive relationship has been reported between high HL and high educational achievement [49]. In our highly educated sample, a total of 68% had university education, compared to 34% in the general Norwegian population [50].

### The association between health literacy and carer burden

We found a statistically significant association between higher HL and lower carer burden. In the literature, low levels of HL have been found to be associated with increased carer burden among family carers of people with other health conditions than dementia [26]. Family carers must often interact with service providers on behalf of the person living with dementia [51]. Such interactions might in themselves increase levels of HL, which then in turn might promote increased access to services that takes some burden off the carer.

Low levels of HL have been associated with reduced service use for chronically ill people [5], poor access to health care among older people [52] as well as for other populations [53],

and reduced likelihood of being able to navigate within healthcare systems [2, 5]. We have shown previously how poorly targeted services may result in increased risks of harm for the person living with dementia and consequently increase carer responsibilities [54].

These studies are consistent with the association we found between lower levels of HL and increased carer burden (B = -0.18 with 95% CI:-0.33,-0.02 p = 0.02). Increase in carer burden was also significantly associated with female gender (B = -8.93 with 95% CI:-12.88, -4.98 p = 0.00). This is consistent with previous studies [55, 56]. Not surprisingly, perhaps, those caring for a person with severe dementia reported heavier carer burden, and this has also been reported elsewhere [41, 57].

## The association between health literacy and health-related quality of life

We found that an increase in HL was associated with an increase in family caregivers' HRQoL as measured by EQvalue (B = 0.003 with 95% CI: 0.001, 0.006 p = 0.04). The association between EQvas and HL was in the same positive direction, but did not reach significance.

Quality of life has been found to be moderately correlated with HL [58]. A large cohort study in UK general practice found that low HL was an independent indicator of poor quality of life among older patients with long-term diseases [59]. However, the use of different instruments complicates comparisons across studies [60].

Few studies have investigated the association between HL and HRQoL in family carers. One study revealed that increased HL was associated with increased quality of life in a population of carers to people with different care needs [9]. Some studies have found positive effects between carer education and carer satisfaction [61]. We found a significant association between higher education and quality of life, for only one of the two HRQoL variables, the EQvas (B = 6.69 with 95% CI:0.51, 12.87, p = 0.03).

Being a male carer was significantly associated with increased EQvalue and EQvas. This is in contrast to a recent systematic review of factors associated with quality of life for family carers of people living with dementia that found no clear association between family carers' HLQoL and their gender [23]. The association between quality of life among family carers' and their age or education was unclear in the systematic review. In our study age was not statistically significantly associated with HRQoL in our study and higher education was statistically significantly associated with HRQoL only when measured with EQvas. Living in urban areas was significantly associated with HRQoL when measured with EQvalue in our study, The systematic review also discussed emerging evidence that the living situation of the care recipient and the family carer may impact carers' quality of life and was not tested in our study. This might be considered a weakness of our analysis.

## The association between health literacy and time spent on informal care

We assumed that higher levels of HL would result in easier access to services or support [19], leading to participants' spending less time themselves on care tasks. Our multiple regression analysis confirmed this, showing significant associations between high HL, and less time spent on informal care (B = -0.03 with 95% CI: -0.06, 0.000, p = 0.046). The RUD questionnaire that informed our study is the most widely used and validated instrument globally, for collecting data regarding informal resource use in dementia care [37, 62, 63]. Nevertheless, our results underscore the fact that there are multiple challenges with measuring time spent on informal care, and the findings from this study should be interpreted carefully. In addition, the high number of missing values in our data for this variable represents a threat to the validity and reliability of this aspect of the overall analysis.

The option of reporting time spent on informal care in a way that allows the total time to exceed the number of available hours could be another limitation for our study. We interpreted the amplification of the number of hours spent on informal care as representing of the fact that carers might carry out several chores simultaneously. Because such an overlap of chores may be present in all cases, we did not exclude the 5 cases that exceeded maximum time.

The option of reporting simultaneous chores could explain why the weekly number of hours spent on care tasks (12.2 in our sample) was considerably higher than the 7.6 hours reported in a Norwegian study of family carers in general [64]. However, studies from the US show that family carers of people living with dementia spend significantly more time on care tasks than other carers, 17.1 hours per week, and 12.5 hours per week, respectively [65].

Our Time variable may therefore best be understood as a respondent rated measure that reflects an *objective* carer burden, similar to how the time variable is interpreted in other studies [66, 67], meaning that more time spent on informal care may indicate more care tasks being performed. The measures of time spent on tasks which were carried out simultaneously were measured equally for all participants. Consequently, it is likely that all participants have reported overlapping tasks when indicating time spent on informal care. It is likely then, that those reporting more hours, actually are providing more care tasks which represents a larger objective burden.

Older age and female gender were significantly associated with more time spent on care. Much of the literature confirms that females participate more in informal care, and spend significantly more time on caregiving for older people living with dementia compared to men [68, 69], but much of the caregiving literature is criticized for reflecting the female perspective [68, 70, 71], which may marginalize, and underestimate male caregiving [72].

It is possible that those caring for someone living in a care home spend less time on some care tasks. We did not include this variable among our pre-specified independent explanatory variables, but a sensitivity analysis did not show significant differences between this group of carers and the rest of the sample. Associations between HL and time spent on informal care in subgroups of family carers for people with dementia could be a direction for future studies.

## Final considerations

In Norway, and in many other countries, policies encourage aging in place, and the delay in moving into long-term residential care. This places as increased reliance on family caregiving to achieve this goal. Despite a growing literature on the role of family carers of older people living with dementia [73–77], and the effects of this role, there is more to learn about how this policy direction influences family carers in the long run [78].

To our knowledge, our study is the first one to measure HL among family carers of older people living with dementia in Norway, and one of the first to investigate the associations between family carers HL and key outcomes that have implications for international lessons. Our key outcomes were measured with validated instruments, although the RUD questionnaire was modified. In our non-probability sample, we noted a bias towards the self-selection of participants with higher levels of HL. Consequently, the results should be treated as exploratory and are not necessarily generalizable to the Norwegian family carer population. The Time variable in particular, should be interpreted carefully because it revealed that carers may add time spent on chores together, even if they are performed simultaneously. Ultimately, the time that family carers spend on informal care should be investigated further with standardized methods and standardized analysis capturing a wider range of care types and time spent on each.

## Conclusion

This study is the first to investigate the role of HL among family carers of older persons living with dementia in Norway, and to investigate whether HL is associated with family carers' level of carer burden, HRQoL, or time spent on informal care. In a sample of 188 family carers with high levels of HL, we found that higher HL was associated with lower carer burden, higher health-related quality of life, and lower time spent by family carers on different care-related tasks.

Viewing the results from this study in the context of earlier studies, it seems that family carers with high levels of HL might be better able to obtain support services that can mitigate negative outcomes. Future studies of HL should seek to obtain a representative sample of family carers to older persons living with dementia, to further investigate these relationships. Increased knowledge in this area would be meaningful to develop fuller understandings of how formal services may support family carers in their role, and support them to increase their HL and thereby potentially enhance their ability to provide sustainable care over time and minimize the burden they sometimes experience.

## Supporting information

**S1 File.**
(SAV)

**S2 File.**
(SPS)

**S3 File.**
(PDF)

**S4 File.**
(PDF)

**S5 File.**
(DOCX)

## Acknowledgments

The authors gratefully acknowledge the 188 family carers who participated in this study, and all of the dementia care coordinators, managers of health services, and other health personnel who provided invaluable help with recruitment.

We also gratefully acknowledge the statistical guidance and support that has been provided by Dr. Jurate Saltyte-Benth, and the expert knowledge about the practical and analytical use of the EQ-5D-5L instrument that was provided by Dr. Kim Rand, both of whom are part of the Health Services Research Unit at Akershus University Hospital.

## Author Contributions

**Conceptualization:** Kristin Häikiö, Jorun Rugkåsa.

**Data curation:** Kristin Häikiö, Jorun Rugkåsa.

**Formal analysis:** Kristin Häikiö, Denise Cloutier, Jorun Rugkåsa.

**Funding acquisition:** Jorun Rugkåsa.

**Investigation:** Kristin Häikiö, Jorun Rugkåsa.

**Methodology:** Kristin Häikiö, Jorun Rugkåsa.

**Project administration:** Kristin Häikiö, Jorun Rugkåsa.

**Resources:** Jorun Rugkåsa.

**Software:** Kristin Häikiö.

**Supervision:** Jorun Rugkåsa.

**Writing – original draft:** Kristin Häikiö.

**Writing – review & editing:** Kristin Häikiö, Denise Cloutier, Jorun Rugkåsa.

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
