## [Decision Letter · Decision Letter 0]

27 Jul 2020

PONE-D-20-05459

Is family carers' Health Literacy associated with carer burden, quality of life, and time spent on informal care for older persons living with dementia?

PLOS ONE

Dear Dr. Häikiö,

Thank you for submitting your manuscript to PLOS ONE. After careful consideration, we feel that it has merit but does not fully meet PLOS ONE’s publication criteria as it currently stands. Therefore, we invite you to submit a revised version of the manuscript that addresses the points raised during the review process.

Each of the reviewers have raised important issues which, when addressed, will make for a stronger article.

We look forward to receiving your revised manuscript.

Kind regards,

Katie MacLure, PhD, MSc (dist)., BSc (1st)

Academic Editor

PLOS ONE

Journal Requirements:

Reviewers' comments:

Reviewer's Responses to Questions

**Comments to the Author**

1. Is the manuscript technically sound, and do the data support the conclusions?

Reviewer #1: Partly

Reviewer #2: Partly

2. Has the statistical analysis been performed appropriately and rigorously? 

Reviewer #1: No

Reviewer #2: Yes

3. Have the authors made all data underlying the findings in their manuscript fully available?

Reviewer #1: Yes

Reviewer #2: Yes

4. Is the manuscript presented in an intelligible fashion and written in standard English?

Reviewer #1: No

Reviewer #2: Yes

5. Review Comments to the Author

Reviewer #1: This study examines the association between health literacy and time spent in caregiving tasks, quality of life and burden in family carers of people living with dementia. This is an interesting question and the existing evidence is used to contexualise the study. While the sample size is adequate for the analysis, the lack of representativeness and the possible bias in a highly educated sample is acknowledged as a limitation.

It would be helpful to list the validated measures in the abstract, along with an indication of direction of associations – i.e. that lower health literacy is associated with higher burden, lower QOL and higher Time.

In the introduction or discussion can you point to any evidence that health literacy can be improved?

I have concerns about using the Time measure. While the RUD is validated, my experience of using it with many carers in a face to face interview is that they really struggle to identify the number of hours they provide as ‘care’ and we usually need to explain what is meant. Completing this via survey is likely to produce unreliable data. I have concerns that three participants reported more hours than possible, the data was exponential and a third of the sample had missing data. Imputing zero hours for these carers (or was it one minute?) seems flawed. If they have zero hours then we wouldn’t consider them a caregiver. It would be more logical to impute the average hours from the rest of the sample, however, this is still flawed when such a large proportion did not report hours. Given these issues, I think the time measure should have been excluded from the analysis. I think you can still report that you used this measure and describe the problems and then discuss in the discussion the difficulty of assessing hours of care through a written questionnaire.

The authors mention that there were a number of skewed variables where medians are shown instead of means. Is it appropriate to perform regression on non-parametric data? Time was log transformed but none of the other variables are mentioned.

In the results and Table 1 there is reference to the HL 6 level scale and 4 level scale as if they were two different scores. However, in the method section it states there was a 6-point Likert scale with 12 items and then the possibility of cutting off scores to make four groups of responders – however, there is no mention of a 6 level and 4 level scale. This needs to be clarified.

For the regression models it is not clear whether any of the variables have been treated as dichotomous variables and what was used as the reference – ie gender – we cannot interpret anything form -.89 in Table 3. Were males or females the reference? It is not explained in the results text either. It is only explained in the discussion.

Could the high levels of health literacy also be due to the self-reported nature of the tool which may lead to people over reporting their HL?

On page 16 you state ‘While we did not find any association between experience as a health professional and HL...’ however, I can’t see that this association was tested in your study.

Editing of expression and language is required throughout, particularly in the use of plurals. Some corrections include:

Page 2 line 25-should be navigate not navigating.

Page 2 line 40 – outcomes not outcome

Page 3 line 54 – do you mean public health view?

Page 6 sentence starting line 101 is very long and complex – could be split up

Page 7 line 143 – developer’s procedures requires a possessive apostrophe

P10 line 193 – statistically not statistical

P12 line 238 delete ‘was’

P13 line 246 delete the full stop

Page 14 - The last sentence needs to be revised.

P15 (no line numbers provided): ‘A positive relationship…’ Not ‘Positive relationship..’; ‘Low levels of HL have been…’ or ‘A low level of HL has been…’; ‘These studies..’ – not ‘This studies..’

P17 – considerably not considerable

Reviewer #2: As the authors note this paper is addressing an under-researched issue for carers of people with dementia. There are a couple of concerns that would need to be addressed:

The sample chosen were of carers where "the care recipient could be living independently, in the same household as the carer or in a nursing home or similar facility." This would not be ocnsistent with at least some of the literature cited where the studies would have dfocused on those living independently or in shared households. There is a literature on family carers of those in nursing homes but it is recognized that their expereince is very different. Also if the person is in a nurisng home then there are staff carers available providing care (perhaps reducing burden) and who access information independently (perhaps meaning family carer health literacy is less important). For the purposes of this study it would have been helpful to control for care setting. Given that those caring at home may also receive interventions from health personnel measn this varoable is not sufficient to capture the effect of location.

Is health literacy associated with carer actions? This is unanswered but potentially health actions or inactions are influencial on the outcomes of interest and this is not considered.

The manuscript would be improved by addressing these issues.

6. PLOS authors have the option to publish the peer review history of their article (what does this mean?). If published, this will include your full peer review and any attached files.

Reviewer #1: No

Reviewer #2: No

---

## [Author Response · Author response to Decision Letter 0]

5 Sep 2020

Dear Dr. MacLure

Thank you for considering publishing our manuscript “Is Health Literacy of family carers associated with carer burden, quality of life, and time spent on informal care for older persons living with dementia?”, and for the reviewers’ comments.. We appreciate the important issues raised by both reviewers to make our manuscript stronger, and have now revised our manuscript accordingly. We have also ensured that our manuscript meets PLOS ONE's style requirements, including those for file naming.

We have uploaded the following documents together with our revised manuscript:

• A responds to each point raised by the academic editor and reviewer(s). This is uploaded as a separate file labeled 'Response to Reviewers'.

• A marked-up copy of our manuscript that highlights changes made to the original version. This is uploaded as a file labeled 'Revised Manuscript with Track Changes'.

• An unmarked version of our revised paper without tracked changes. This is uploaded as a file labeled 'Manuscript'.

• Additional information regarding the survey and questionnaires used in the study. We have provided the questionnaire as applied in Norwegian, and English versions of the scales/questions. This is uploaded as ‘Supporting Information’

• A minimal dataset from where our analysis and results can be replicated together with our syntax file for the regression analysis. Uploaded as ‘supporting Information’.

We have made the following alterations to the manuscript in response to the reviews:

Reviewer #1: 

1. It would be helpful to list the validated measures in the abstract, along with an indication of direction of associations – i.e. that lower health literacy is associated with higher burden, lower QOL and higher Time.

Author’s reply:

• Thank you for your advice. The validated measures and the indication of direction of associations are now included in the abstract. See page 2, line 31-34 and line 44-50.

2. In the introduction or discussion can you point to any evidence that health literacy can be improved?

Author’s reply:

• Thank you for pointing this out. On page 4, line 72-74 the following text is included, in addition to a reference: “A systematic review aimed at establishing the efficacy of interventions to improve health literacy and health behavior, concluded that HL interventions hold potential to combat health inequalities [2,7,8].”

3. I have concerns about using the Time measure. While the RUD is validated, my experience of using it with many carers in a face to face interview is that they really struggle to identify the number of hours they provide as ‘care’ and we usually need to explain what is meant. Completing this via survey is likely to produce unreliable data. I have concerns that three participants reported more hours than possible, the data was exponential and a third of the sample had missing data. Imputing zero hours for these carers (or was it one minute?) seems flawed. If they have zero hours then we wouldn’t consider them a caregiver. It would be more logical to impute the average hours from the rest of the sample, however, this is still flawed when such a large proportion did not report hours. Given these issues, I think the time measure should have been excluded from the analysis. I think you can still report that you used this measure and describe the problems and then discuss in the discussion the difficulty of assessing hours of care through a written questionnaire.

Author’s reply:

• We share the reviewer’s concern regarding the difficulties of using the RUD questionnaire in a self-administered survey, and agree in general with the concern for reliability and validity of the Time-variable. We have now made this clearer for the reader in the discussion and made these issues more transparent. See page 19-20, line 391-414:

”The RUD questionnaire that informed our study is the most widely used and validated instrument globally, for collecting data regarding informal resource use in dementia care [37,62,63]. Nevertheless, our results underscore the fact that there are multiple challenges with measuring time spent on informal care, and the findings from this study should be interpreted carefully. In addition, the high number of missing values in our data for this variable represents a threat to the validity and reliability of this aspect of the overall analysis. 

The option of reporting time spent on informal care in a way that allows the total time to exceed the number of available hours could be another limitation for our study. We interpreted the amplification of the number of hours spent on informal care as representing of the fact that carers might carry out several chores simultaneously. Because such an overlap of chores may be present in all cases, we did not exclude the 5 cases that exceeded maximum time. 

The option of reporting simultaneous chores could explain why the weekly number of hours spent on care tasks (12.2 in our sample) was considerably higher than the 7.6 hours reported in a Norwegian study of family carers in general [64]. However, studies from the US show that family carers of people living with dementia spend significantly more time on care tasks than other carers, 17.1 hours per week, and 12.5 hours per week, respectively [65]. 

Our Time variable may therefore best be understood as a respondent rated measure that reflects an objective carer burden, similar to how the time variable is interpreted in other studies [66,67], meaning that more time spent on informal care may indicate more care tasks being performed. The measures of time spent on tasks which were carried out simultaneously were measured equally for all participants. Consequently, it is likely that all participants have reported overlapping tasks when indicating time spent on informal care. It is likely then, that those reporting more hours, actually are providing more care tasks which represents a larger objective burden.” 

• To further explain the time variable we revised the text on page 9, line 189-192: “The time spent on each of the five task clusters were calculated by multiplying the “hours spent on a typical care day” with the “number of days” in which this task was carried out during the last 30 days. The Time variable is calculated by adding up the time spent on all task clusters.”

• Instead of removing the Time-variable, we present the results with a greater emphasis on careful interpretation and discuss the low reliability and validity of this variable. As the reviewer also pointed out, there are great issues with most (perhaps all) measures of time spent on informal care. By publishing our analysis and results we contribute to study the association between health literacy (HL) and time spent on informal care on populations of family carers to older people living with dementia. To make it clearer to the readers how time was calculated we also included this text on page 9, line 189-192: “The time spent on each of the five task clusters were calculated by multiplying the “hours spent on a typical care day” with the “number of days” in which this task was carried out during the last 30 days. The Time variable is calculated by adding up the time spent on all task clusters.”

• The comment regarding the use of imputation on the Time variable was very helpful. We have changed our procedure of imputation to include mean value where possible. This is explained in detail on page 11, line 233-244: “Imputation was undertaken for missing values for the following three of the dependent variables: HL, carer burden and Time. 

For the HL-variable and the carer burden variable; missing values were replaced with a mean value [42,43]. In those cases were imputation was undertaken, fewer than 3 values were missing. 

For the Time variable imputation was undertaken according to the following two procedures: 1) In the few cases where the respondent had answered only one of the two questions used to calculate time spent on informal care, either “hours spent”, or “number of days” we used imputation of the mean value for the sample. 2) When both values in a cluster of tasks were missing we used the value zero. This occurred in 32 cases. While this included the risk of underestimating the time spent on informal care, it reduced the risk of Type 1 errors in the regression analysis. To test whether this procedure increased the risk of Type II errors, we conducted a sensitivity analysis with imputed mean values.”.

This did not change the results significantly from previous results regarding the association between HL and the outcome variables, but it did slightly change which of the explanatory independent variables that reached the significance level. Because of this we have revised the text in page 20 line 415-419:”Older age and female gender were significantly associated with more time spent on care. Much of the literature confirms that females participate more in informal care, and spend significantly more time on caregiving for older people living with dementia compared to men[68], but much of the caregiving literature is criticized for reflecting the female perspective [71,72], which may marginalize, and underestimate male caregiving [73].”

• The use of imputation of mean, when possible, resulted in 5 cases exceeding maximum hours per month, rather than three. We did not think it was appropriate to exclude these cases, but we have included a sensitivity analysis where we did this, as presented on page 15, line 308-310: “Five participants indicated having spent time exceeding the number of hours in a month (>720. Therefore, we ran the regression analyses of Time again, excluding these five cases. HL remained statistically significantly associated with Time (B -0.29 with 95% CI: -0.06,0.000, p=0.049)“. 

• As the number of participants exceeding the maximum time per month, the manuscript is revised accordingly on page 19, line 397-401: “The option of reporting time spent on informal care in a way that allows the total time to exceed the number of available hours could be another limitation for our study. We interpreted the amplification of the number of hours spent on informal care as representing of the fact that carers might carry out several chores simultaneously. Because such an overlap of chores may be present in all cases, we did not exclude the 5 cases that exceeded maximum time.”. We also added the following text on page 19, line 402-414: “The option of reporting time spent on informal care in a way that allows the total time to exceed the number of available hours could be another limitation for our study. We interpreted the amplification of the number of hours spent on informal care as representing of the fact that carers might carry out several chores simultaneously. Because such an overlap of chores may be present in all cases, we did not exclude the 5 cases that exceeded maximum time.”. On page 20, line 408-419 we added: “Our Time variable may therefore best be understood as a respondent rated measure that reflects an objective carer burden, similar to how the time variable is interpreted in other studies [66,67], meaning that more time spent on informal care may indicate more care tasks being performed. The measures of time spent on tasks which were carried out simultaneously were measured equally for all participants. Consequently, it is likely that all participants have reported overlapping tasks when indicating time spent on informal care. It is likely then, that those reporting more hours, actually are providing more care tasks which represents a larger objective burden.”

• We did not think it was appropriate to impute means were all values were missing in a cluster of care tasks. However, we have included a sensitivity analysis where we did this, as presented on page 15, line 305-317: “In 32 cases all values were missing in a cluster for the Time variable. The sensitivity analysis where we used imputed means instead of zero in these cases showed that a higher level of HL was still statistically significantly associated with less time spent on informal care (B -0.33 with 95% CI: -0.07, 0.000, p=0.049)“. The sensitivity analysis did not change the results significantly for the association between HL and the outcome variables.

4. The authors mention that there were a number of skewed variables where medians are shown instead of means. Is it appropriate to perform regression on non-parametric data? Time was log transformed but none of the other variables are mentioned.

Author’s reply:

• We now explain this better in the manuscript. The distributions of the values in some of the variables were skewed. However, the assumption for regression analysis is that the residuals should be normally distributed. We have included this information on page 12, line 251-252: “All variables except the Time variable had normally distributed residuals.” 

• On page 12, line 251-254 we included the following text, as a response to the reviewer’s comment on log-transformation of variables: “All variables except the Time variable had normally distributed residuals. We therefore used log time transformation [42] to meet the assumptions of linearity of the residuals. One minute was added to the total time for each participant in order to avoid logarithms of zero for these calculations.”

5. In the results and Table 1 there is reference to the HL 6 level scale and 4 level scale as if they were two different scores. However, in the method section it states there was a 6-point Likert scale with 12 items and then the possibility of cutting off scores to make four groups of responders – however, there is no mention of a 6 level and 4 level scale. This needs to be clarified.

Author’s reply:

• We agree that this was unclear in our manuscript. We have now clarified this in our manuscript by adding the following text in the Measures section on page 8 line 160-162: “To represent the level of HL as descriptive categories, we converted the 6 levels to 4 as suggested by the scale’s developers, and we followed their procedures for calculating cut-off values [33].”. We also added this text on page 8, line 165-166:” The 6-level measure was used in the regression analysis, and the 4-level measure in the descriptive analyses.”

6. For the regression models it is not clear whether any of the variables have been treated as dichotomous variables and what was used as the reference – ie gender – we cannot interpret anything form -.89 in Table 3. Were males or females the reference? It is not explained in the results text either. It is only explained in the discussion.

Author’s reply:

• We apologize for not making this clear in Table 3. We have now included information in the table that female is the reference. We added the following explanations to Table 3, on page 15: “ aThe Time variable is the log transformed variable. bVariables are binary and coded no=0/yes=1, cThe variable Dementia severity is coded mild dementia=0/severe dementia=1.”

This was described in the Measures-section but we agree that this should be clearly stated in Table 3.

7. Could the high levels of health literacy also be due to the self-reported nature of the tool which may lead to people over reporting their HL?

Author’s reply:

• We agree that it is likely that the willingness to participate in a self-reported survey may be greater among those with higher HL. We revised the text on page 16, line 330-336: “Also, there may be selection bias in the participants in this study, such that those who chose to participate had higher levels of HL than the general population [45-46]. A self-administered survey makes cognitive demands on participants [45], which might impact who volunteered to take part [46]. It may be likely that the combination of a non-probability sampling method, voluntarily participation of family carers, the self-reported nature of the HL-scale, and self-administration of the survey, have together contributed to participants with higher levels of HL.”

8. On page 16 you state ‘While we did not find any association between experience as a health professional and HL...’ however, I can’t see that this association was tested in your study.

Author’s reply:

• We agree with the reviewer and removed this from the paper.

9. Editing of expression and language is required throughout, particularly in the use of plurals. Some corrections include:

Page 2 line 25-should be navigate not navigating.

Page 2 line 40 – outcomes not outcome

Page 3 line 54 – do you mean public health view?

Page 6 sentence starting line 101 is very long and complex – could be split up

Page 7 line 143 – developer’s procedures requires a possessive apostrophe

P10 line 193 – statistically not statistical

P12 line 238 delete ‘was’

P13 line 246 delete the full stop

Page 14 - The last sentence needs to be revised.

P15 (no line numbers provided): ‘A positive relationship…’ Not ‘Positive relationship..’; ‘Low levels of HL have been…’ or ‘A low level of HL has been…’; ‘These studies..’ – not ‘This studies..’

P17 – considerably not considerable

Author’s reply:

• Our apologies for the misspellings. The misspellings that are pointed out by the reviewer are now corrected. 

Reviewer #2: 

1. The sample chosen were of carers where "the care recipient could be living independently, in the same household as the carer or in a nursing home or similar facility." This would not be consistent with at least some of the literature cited where the studies would have focused on those living independently or in shared households. There is a literature on family carers of those in nursing homes but it is recognized that their experience is very different. Also if the person is in a nursing home then there are staff carers available providing care (perhaps reducing burden) and who access information independently (perhaps meaning family carer health literacy is less important). For the purposes of this study it would have been helpful to control for care setting. Given that those caring at home may also receive interventions from health personnel means this variable is not sufficient to capture the effect of location.

Author’s reply:

• This is an interesting comment. Living situation was not one of our pre-defined independent explanatory factor. The independent variables were selected from theory and initial correlation analysis. Whether the care-recipient was living in a nursing home, living independently or in shared household was not one of the variables that had a strong correlation to the outcome variables. However, in light of the reviewer’s comment we decided to include a sensitivity analysis where those caring for someone living in a nursing home or similar are compared with the remaining sample.

We have added this on page 15-16, line 315-321: “It is possible that the large differences in time spent on informal care could be related to whether the care recipient lived in a nursing home, which might reduce the need for carer input. We conducted a sensitivity analysis on the time variable using independent sample T-test to investigate difference of mean between those who cared for a person living in a nursing home and the remaining respondents. We found no statistically significant difference (mean 3.6 hours, SD 2.3 vs .2 hours, SD 2.6, p = 0.31). The correlation between the time variable and caring for a person living in nursing home was also not significant (Pearson’s R = -0.14, p=0.051).”

We also added this text to page 10, line 213-215:” Information about whether the care recipient was living in a nursing home facility or not was collected and the variable was coded as: living in a nursing home facility (no=0/yes=1).”

2. Is health literacy associated with carer actions? This is unanswered but potentially health actions or inactions are influential on the outcomes of interest and this is not considered.

Author’s reply:

• We are not entirely sure if we understand this comment. Implicit in the concept of health literacy, as explained in the Introduction, is a presumption that someone’s level of health literacy will shape their (health related) actions and behaviors. Our analysis is based on this presumption and aimed to explore whether this then impacts key outcomes. We did not measure any ‘carer actions’ that can be included in the analysis. 

Additional improvements made to the manuscript by the authors:

To better explain how we had modified the RUD questionnaire we added the following on page 9, line 183-186: “This included adding one more traditionally male chore; maintenance of the house, to make the measure more relevant to male carers. We also added time spent talking to the participant on the phone and time spent interacting with health personnel.” 

To improve the manuscript further we reorganized the text on page 14, line 290-301.

Because the results changed slightly when using imputation of mean, we revised the text in the discussion according to the new results, see track changes in Discussion section.

In light of the grammatical errors pointed out by reviewer 1, and that several of the reviewers’ comments related to the clarity of the manuscript, we conducted a careful edit of the entire manuscript to correct further errors and enhance clarity. 

Best regards, 

On behalf of the authors

Kristin Häikiö

---

## [Decision Letter · Decision Letter 1]

26 Oct 2020

Is Health Literacy of family carers associated with carer burden, quality of life, and time spent on informal care for older persons living with dementia?

PONE-D-20-05459R1

Dear Dr. Häikiö,

We’re pleased to inform you that your manuscript has been judged scientifically suitable for publication and will be formally accepted for publication once it meets all outstanding technical requirements.

Kind regards,

Katie MacLure, PhD, MSc (dist)., BSc (1st)

Academic Editor

PLOS ONE

Additional Editor Comments (optional):

Reviewers' comments:

Reviewer's Responses to Questions

**Comments to the Author**

1. If the authors have adequately addressed your comments raised in a previous round of review and you feel that this manuscript is now acceptable for publication, you may indicate that here to bypass the “Comments to the Author” section, enter your conflict of interest statement in the “Confidential to Editor” section, and submit your "Accept" recommendation.

Reviewer #2: All comments have been addressed

2. Is the manuscript technically sound, and do the data support the conclusions?

Reviewer #2: Yes

3. Has the statistical analysis been performed appropriately and rigorously? 

Reviewer #2: Yes

4. Have the authors made all data underlying the findings in their manuscript fully available?

Reviewer #2: Yes

5. Is the manuscript presented in an intelligible fashion and written in standard English?

Reviewer #2: Yes

6. Review Comments to the Author

Reviewer #2: all comments have been addressed. This will be an interesteing addition to the literature and should encurage additional work

7. PLOS authors have the option to publish the peer review history of their article (what does this mean?). If published, this will include your full peer review and any attached files.

Reviewer #2: No

---

## [Editor Report · Acceptance letter]

30 Oct 2020

PONE-D-20-05459R1 

Is Health Literacy of family carers associated with carer burden, quality of life, and time spent on informal care for older persons living with dementia? 

Dear Dr. Häikiö:

I'm pleased to inform you that your manuscript has been deemed suitable for publication in PLOS ONE. Congratulations! Your manuscript is now with our production department. 

Kind regards, 

on behalf of

Dr. Katie MacLure 

Academic Editor

PLOS ONE